# Ergonomics of E-Learning Workstations and the Prevalence of Musculoskeletal Disorders—Study among University Students

**DOI:** 10.3390/ijerph20043309

**Published:** 2023-02-14

**Authors:** Magdalena Janc, Zbigniew Jozwiak, Agnieszka Jankowska, Teresa Makowiec-Dabrowska, Jolanta Kujawa, Kinga Polanska

**Affiliations:** 1Department of Environmental and Occupational Health Hazards, Nofer Institute of Occupational Medicine (NIOM), 91-348 Lodz, Poland; 2Department of Medical Rehabilitation, Faculty of Health Sciences, Medical University of Lodz, 92-213 Lodz, Poland

**Keywords:** COVID-19 pandemic, ergonomic, musculoskeletal symptoms, remote learning, ROSA method

## Abstract

The prevalence of musculoskeletal disorders (MSD) has increased significantly in recent years. The COVID-19 pandemic has led to a fundamental change in the lifestyles, ways of learning and working patterns of the general population, which in turn, might lead to health consequences. The aim of this study was to evaluate the conditions of e-learning and the impact of the learning modality on the occurrence of musculoskeletal symptoms among university students in Poland. This cross-sectional study included 914 students who completed an anonymous questionnaire. The questions covered two time periods (before and during the COVID-19 pandemic) and were aimed at obtaining information about lifestyle (including physical activity using the modified International Physical Activity Questionnaire, 2007 (IPAQ), perceived stress and sleep patterns), the ergonomics of computer workstations (by Rapid Office Strain Assessment, 2012 (ROSA) method), the incidence and severity of musculoskeletal symptoms (by the Nordic Musculoskeletal Questionnaire, 2018 (NMQ)) and headaches. The main differences between the two periods were statistically significant according to the Wilcoxon test in terms of physical activity, computer use time, and severity of headaches. During the COVID-19 pandemic, there was a significant increase in MSD (68.2% vs. 74.6%) and their intensity (2.83 ± 2.36 vs. 3.50 ± 2.79 points) among the student population (*p* < 0.001). In the group of students with MSD, there was a high musculoskeletal load, due to the lack of ergonomic remote learning workstations. In future, a thorough study should be carried out, and there is an urgent need to raise students’ awareness of arranging learning workstations according to ergonomic principles in order to prevent the occurrence of musculoskeletal problems.

## 1. Introduction

The prevalence of work-related musculoskeletal disorders (WMSD) has increased significantly in recent decades. Static as well as dynamic physical loading are the most important causes of absence and disability at work [1]. The World Health Organization (WHO) has described them as multifactorial, involving physical, psychosocial, individual, organizational and socio-cultural factors. Performing tasks using a computer and in a sedentary position for two-thirds of the working time increases the risk of many chronic diseases. Nowadays, it is not possible to limit work with screen monitors, but the position and the way of doing this work should be rationalized. Employees are guaranteed assistance in organizing an ergonomic workstation by their employers. Legal considerations, however, do not apply in the case of students who have been forced to change their studying environment and organization due to the COVID-19 pandemic situation.

The COVID-19 pandemic has led to a fundamental change in the lifestyle of the general population. In February 2020, the coronavirus (SARS-CoV-2) infection causing severe acute respiratory syndrome (SARS) was named COVID-19 disease [2]. As part of the campaign against the virus, restrictions were introduced all over the world. In order to limit the transmission of the virus, recommendations were issued including restrictions on mobility and face-to-face contact which made it possible or obligatory to carry out work duties at home. These restrictions also applied to schoolchildren as well as university students [3,4,5].

Although the strategy of isolation and restrictions on human contact has proved effective in controlling infection rates, it has significantly affected many important aspects of the health of the restricted populations. Some research has assessed changes in lifestyle and the consequences for people’s health state and well-being including ailments and the level of stress [6,7]. Other studies were concerned the evaluation of home conditions, including technical aspects such as internet speed and hardware facilities, pointing to the factors that may predispose to the occurrence of negative health effects [8,9,10,11,12,13,14,15,16,17,18,19,20]. Although there are many studies on the impact of COVID-19 pandemic on learning, lifestyle, and health outcomes, most often they focus on a single aspect such as student’s attitudes and perceptions of e-learning or its impact on MSD. In our study we approached the problem in a comprehensive way by investigating the ergonomics of students’ computer-based workstations (using internationally validated/standardized methods which take into account not only the theoretical but also the practical adaptation of equipment) and evaluating physical activity, musculoskeletal symptoms, headaches and sleep duration. The aim of this study was to evaluate the conditions of e-learning and the impact of the learning modality on the occurrence of musculoskeletal symptoms. The specific objectives were to assess the ergonomics of new e-learning workstations, and the frequency and level of student complaints and lifestyle changes before and during the COVID-19 pandemic.

## 2. Materials and Methods

### 2.1. Study Design

This cross-sectional study was conducted in Poland. The invitation to participate in the survey was sent to three universities in Lodz, namely, the University of Lodz, the Medical University of Lodz and the Technical University of Lodz. The survey involved the completion of an anonymous, online questionnaire (Google forms). The questions covered two time periods, i.e., before the COVID-19 pandemic and during the period from October 2020 to June 2021 (hereafter referred to as the pandemic period). The pandemic period was selected to cover full academic year and to exclude the period of adaptation to the pandemic situation and the full lockdown. Most of the courses within the pandemic period were performed remotely (or alternatively in hybrid form). The data for both periods was obtained retrospectively.

### 2.2. Participants

The criterion for inclusion in the study was student status at one of the three universities. Fully completed questionnaires that were included in the analyses were obtained from 914 respondents.

### 2.3. Instruments

The questionnaire consisted of four parts. Part one included sociodemographic questions such as age and sex. In addition, height, weight and the name of university were obtained. 

The second part of the questionnaire was designed to collect information about physical activity using the modified International Physical Activity Questionnaire (IPAQ)—short version [21,22]. It is based on seven questions about the type of physical activity and its duration. International guidelines for physical activity require it to be reported in terms of the metabolic equivalent (MET), which corresponds to the oxygen uptake at rest in a seated position for 1 min (approximately 3.5 mL oxygen/min/kg body mass). In the case of vigorous physical activity, a value of 8 METs was assigned, for moderate physical activity 4 METs and for low physical activity (walking) 3.3 METs were considered. In order to determine the total amount of physical activity undertaken, the time spent doing the activity was multiplied by the corresponding factor. The students referred to the distribution of a typical week before the COVID-19 pandemic and to the period between October 2020 to June 2021. The analyses assessed whether students followed the WHO guidelines, which set a minimum amount of physical activity per week of 150–300 min for moderate aerobic physical activity and at least 75–150 min for vigorous aerobic physical activity or an equivalent combination of moderate and vigorous intensity activity [23].

The Rapid Office Strain Assessment (ROSA) method [24,25,26] was used for assessment of ergonomic of computer workstations. The method measures the quality and use of all the equipment and organizational aspects of the workstations which have a direct impact on the body position adopted during work. Using an appropriate scoring system, the height and depth of the seat, and the positioning of armrests and backrest are assessed, which represent Section A on a scale of 2–9 points. Then the positioning of the monitor and telephone (Section B on a scale of 1–9 points), the keyboard and mouse (Section C on a scale of 1–9 points) and the monitor and accessories (Section D on a scale of 1–9 points) are assessed. A higher number of points indicates less ergonomic workstations. The overall ROSA score (on a scale of 1–10 points) is obtained by reading from the table a summary value that is a combination of the scores of Sections A and D, taking into account the duration of use of the separate components of the computer equipment. Score values of 1–2 points indicate low, 3–4 points moderate, 5–7 points high and 8–10 points very high musculoskeletal load and risk of discomfort. These values indicate when and the extent to which action to avoid the negative effects of non-ergonomic workstations. The higher the score, the greater the strain on the musculoskeletal system and the risk of disorders. In the analysis using the ROSA method, the period before the pandemic was omitted, due to the lack of a classic computer workstation during desktop learning resulting from the diversity of activities. Several questions have been added regarding organization, study time and the type of device most commonly used for e-learning.

The last part of the questionnaire focused on the incidence and severity of musculoskeletal symptoms, headaches, perceived stress and sleep patterns. The musculoskeletal symptoms were assessed by selected items of the Nordic Musculoskeletal Questionnaire (NMQ) [27,28,29]. The questions relate to the occurrence of symptoms in different areas of the body and at different time points. In order to determine the impact of the pandemic period on the occurrence of musculoskeletal complaints, the respondents were asked to tick all complaints occurring before the pandemic and the predominant ones occurring during the COVID-19 pandemic period. The onset of headaches, the level of pain and the period of the day when it occurred were also considered. To assess the severity of musculoskeletal symptoms and headaches, the Visual Analogue Scale (VAS) [30] was used, on which the level of discomfort is marked on a 10-point scale, where 0 means no symptoms and 10 is the maximum value.

The level of perceived stress was assessed based on a 5-point scale from 5 (very high) to 1 (very low), with 3 considered as moderate. To determine the relationship between the perceived level of stress and the occurrence of musculoskeletal complaints, a cutoff of 3 and higher was set. The sleep aspect was analysed in terms of the time it was taken and its length.

### 2.4. Statistics

Basic statistics, including the mean and standard deviation were used to access differences between the two periods. The results were considered as statistically significant at *p* < 0.05 value according to the Wilcoxon paired test. In order to assess the associations between the occurrence of musculoskeletal complaints in the neck region or lumbar spine and workplace characteristics and lifestyle factors, we tested whether their occurrence or severity differed significantly in those with and without complaints. A one-way analysis of variance (ANOVA) was used here. The program STATISTICA software, v.13.3, StatSoft, Tulsa, OK, USA was used for the analyses.

## 3. Results

### 3.1. Chracteristics of the Study Population

Females accounted for 64% of the study population and the mean age of participants was 21.7 ± 2.2 years. No significant change in body weight was observed (66.8 kg ± 14.5 kg vs. 68.0 kg ± 15.3 kg; *p* > 0.05).

### 3.2. Comparison of the Variables Studied in the Periods before and during the COVID-19 Pandemic

Significant differences were found in the levels of vigorous, moderate and walking physical activity undertaken, which translate into a significant reduction in the number of subjects following WHO recommendations from 70% to 62% during the pandemic (*p* < 0.001) (Table 1). During the compared periods, an increase in computer use time was noted (from 8.1 h ± 5.2 h to 11.4 h ± 4.6 h; *p* < 0.001). Moreover, the proportion of students taking a break less than once per hour increased (from 34% to 39%; *p* < 0.001).

Across the group, the results of each section and the overall ROSA assessment revealed a medium musculoskeletal load of 3.6 ± 1.5 points. The most common equipment used for e-learning was a laptop (80%), followed by a desktop computer (16%) and a smartphone (2%). 

There was a significant increase in the percentage of participants who experienced musculoskeletal complaints (68% vs. 75%) and their intensity (2.8 ± 2.4 vs. 3.5 ± 2.8 points), for the periods before and during the pandemic, respectively (*p* < 0.001). Musculoskeletal problems during the pandemic period were most common for the neck and lumbar regions (Figure 1).

Moderate or higher levels of stress were reported by more than 80% of students, regardless of the study period (*p* > 0.05) (Table 1). During the pandemic, the number of headache sufferers as well as the level of severity of headaches increased from 71% to 78% (*p* < 0.001) and from 3.1 ± 2.6 to 3.8 ± 4.2 points (*p* < 0.001), respectively. The percentage of subjects going to bed after midnight increased significantly in the second period from 45% to 54% (*p* < 0.001), but sleep duration did not change significantly on either weekdays or weekends (*p* > 0.05).

### 3.3. Factors That Modify the Occurrence of MSD Symptoms

The intensity of physical activity was a consistent differentiating factor between those with and without musculoskeletal symptoms (Table 2 and Table 3). Before the pandemic, vigorous physical activity was statistically higher in those without symptoms in the lumbar region (*p* < 0.05) (Table 2). During the pandemic, the levels of walking and moderate PA were lower for people without musculoskeletal symptoms (*p* < 0.05) (Table 3). 

A comparison of the ROSA scores of computer workstations with the neck and lumbar MSD showed that those with symptoms had worse computer workstations. The ROSA final score was 4.0 ±1.5 points in the group of students with neck and lumbar complaints and 3.8 ±1.4 points in the group not reporting such symptoms (*p* < 0.01). The sections A (chair) and D (monitor + additional equipment) scored higher among students with MSD compared with the group without symptoms (*p* < 0.05). 

Students without complaints slept longer (30 min before the pandemic) and had 5% lower stress levels. Before the pandemic, they experienced headaches with less frequency when combined with neck or lumbar complaints (*p* < 0.05). During e-learning, the incidence of headaches in participants with musculoskeletal complaints in the neck area was significantly higher (*p* < 0.001). Students with complaints in the lumbar area also reported an increase in headaches (*p* < 0.01). The presence of stress was 7% higher in those with MSD complaints, both of the neck and lumbar region (*p* < 0.05).

## 4. Discussion

The COVID-19 pandemic has certainly changed many aspects of our lives. One of the changes was the necessity of implementing of e-learning. It has become a challenge for academics as well as students, both in the area of workstation/study organization and lifestyle habits. Inadequate organization of the workstation and insufficient physical activity level might have certain health consequences. It is alarming that as many as 68% (before the pandemic) and 75% (during the pandemic) of the students have experienced MSD given that these are young people who have not yet started their working life. Hendi et al. in 2019 found that 65% of health specialty students reported musculoskeletal ailments before the pandemic [31]. A high percentage was also reported among dental, physiotherapy and music students [32,33,34]. However, in these specialties, the determining factor in the occurrence of musculoskeletal complaints was related to forced body positions and repetitive movements. The incidence and severity of MSD increased during the pandemic, as also reported by other authors. In a study of an Italian student population, Roggio et al. reported that 73% of the study group experienced pain, particularly in the neck and low back [35]. 

While the occurrence of some of the complaints before the pandemic could probably be attributed to sports or traffic injuries, the occurrence of such a number in a single year of study/e-learning with some pandemic restrictions is worrisome. The predominant areas of the body for such complaints are the neck and lumbar spine, which indicates their associations with prolonged sitting positions and working at computer workstations. The very long time of computer use noted in our study is due to the fact that we considered total time. i.e., computer usage for study and during leisure time activities. This measure was justified by the need to take into account the impact of the actual time spent at the computer on the occurrence MSD [12,13,14,15,16,17,18,19,20,36,37]. However, prolonged computer use, was not a differentiating factor in the occurrence of musculoskeletal complaints of the neck and lumbar areas. It should be noted here that the style of participation in activities did not necessitate continuous maintenance of a sitting position. Students adopted a variety of positions while e-studying, including lying forward or backward on a couch/floor or sitting on a kitchen bar chair or a pouffe [13]. As the strenuousness of working with a computer is determined not only by the overall time spent using it, but also by the uninterrupted time spent on working, the respondents were asked after what time they take a break. The data showed an increase in the percentage of students taking a break less than once per hour during the pandemic period, but students’ breaks depended on the duration of the lesson unit for which they were scheduled.

The computer workstation was another factor determining the risk of musculoskeletal complaints during the pandemic. This applies to both their specific computer components and the pattern of their use. The average of the final evaluation using the ROSA method indicated a medium musculoskeletal load for the entire group. However, analyses for particular symptoms suggested a higher level of load, i.e., a low ergonomic position or low functional use of computer workstations. For e-studying, the most common device used was a laptop, which, as a stand-alone device, is not suitable for long-term work [12]. It forces the adoption of a flexion position in the cervical region, which leads to hyper tension in the neck muscles during prolonged work [14]. A factor affecting headache is the presence of musculoskeletal complaints in the neck area in the form of increased muscle tension, and perceived levels of stress can also exacerbate excessive tension in this area, causing an increase in tension-type headaches. A very high percentage of students (>80%) reported feeling stress, both before and during the pandemic [38]. However, experiencing stress at or above moderate levels was higher for students with complaints in both the cervical and lumbar regions. 

The advantages of this survey are the large sample size and the comprehensive assessment of parameters related to lifestyle (physical activity, sleep), to the ergonomics of computer workstations and to health (in terms of musculoskeletal complaints and headaches). It should be noted that the study used internationally validated/standardized questionnaires/tools, including the evaluation of not only theoretical but also practical adaptation of computer workstations. Some limitations of the study may be the cross-sectional study design and the fact that the data was collected retrospectively. Moreover, the subjective evaluation of the analysed parameters needs to be underlined. Thus, bias resulting from possible alterations in the perception of the activities carried out and the perception of health outcomes (MSD, headaches) as well as the influence attributable to social isolation cannot be excluded. The study did not capture all possible factors that could affect the occurrence of MSD; nevertheless, it is known that the complaints were mostly related to the physical activity undertaken and non-ergonomic learning positions and workstations. We recommend to conduct obligatory training on ergonomics issues. It is also important to conduct further studies after a longer period of time to evaluate the strength of the impact of changes forced by the COVID-19 pandemic. The results of this study can be applicable to other student populations; however, the level of awareness and available equipment may vary from country to country.

## 5. Conclusions

During the COVID-19 pandemic, there was an increase in musculoskeletal complaints among the student population. In the group of students with MSD complaints, there was a high musculoskeletal load, due to the lack of ergonomic remote learning workstations. A thorough study should be carried out in future, and changes are needed according to the recommendations of the ROSA method.

## Figures and Tables

**Figure 1 ijerph-20-03309-f001:**
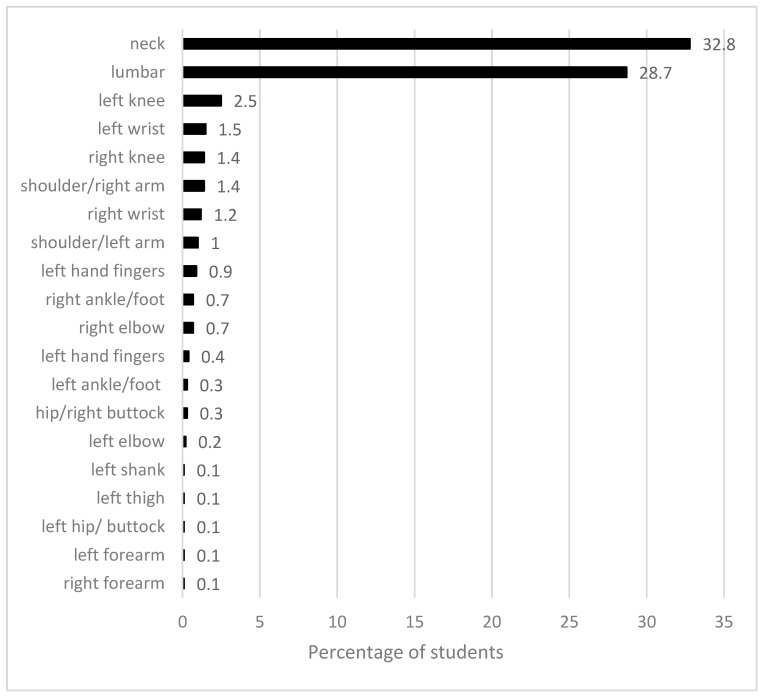
Location of dominant musculoskeletal complaints during the COVID-19 pandemic.

**Table 1 ijerph-20-03309-t001:** Differences between the compared periods.

Variables	Before COVID-19 Pandemic		During COVID-19 Pandemic	
	M ± SD	Percentage of Participants (%)	M ± SD	Percentage of Participants (%)
Vigorous PA (8MET·minutes·days per week)	579.3 ± 1737.5		567.1 ± 1763.3 ***	
Moderate PA (4MET·minutes·days per week)	829.9 ± 2508.8		614.3 ± 1040.0 ***	
Walking (3.3MET·minutes·days per week)	682.3 ± 1008.9		676.1 ± 1127.9 ***	
Following WHO standards for PA		70.0		62.3 ***
Computer use time (hours/day)	8.1 ± 5.2		11.4 ± 4.6 ***	
Breaks less than once per hour		33.8		39.2 ***
Section A (chair) (points)	N/S		3.2 ± 1.5	
Section B (monitor + phone) (points)	N/S		2.3 ± 0.9	
Section C (keyboard + mouse) (points)	N/S		2.4 ± 1.3	
Section D (monitor +additional equipment) (points)	N/S		2.8 ± 1.2	
ROSA (final score) (points)	N/S		3.6 ± 1.5	
No MSD symptoms		31.8		25.4 ***
Intensity of MSD complaints (points)	2.8 ± 2.4		3.5 ± 2.8 ***	
Stress (moderate or higher)		82.0		81.8
Severity of headache (points)	3.1 ± 2.6		3.8 ± 4.2 ***	
Headache (yes)		71.4		78.0 ***
Onset of sleep after midnight (weekdays)		44.5		53.7 ***
Onset of sleep after midnight (weekend)		72.3		73.8
Sleep duration on study days (hours)	6.7 ± 1.2		7.0 ± 1.5	
Sleep duration on non-study days (hours)	8.7 ± 1.6		8.8 ± 1.7	

M—mean; SD—standard deviation; PA—physical activity; N/S—not stated; *** *p* < 0.001.

**Table 2 ijerph-20-03309-t002:** Comparison of lifestyle parameters and health aspects in subgroups of people with and without complaints in the neck region or lumbar spine before the pandemic (only statistically significant differences are presented).

	Neck				Lumbar			
	M ± SD = 0	M ± SD = 1	% = 0	% = 1	M ± SD = 0	M ± SD = 1	% = 0	% = 1
Vigorous PA (8MET·minutes·days per week)					1033.6 ± 1207.2	830.9 ± 1180.2 *		
Stress (moderate or higher)			79.5	85.1 ***			79.5	85.0 *
Severity of headache (points)	2.5 ± 2.4	3.4 ± 2.5 ***			3.1 ± 0.9	3.6 ± 2.6 ***		
Headache (yes)			63.1	81.3 ***			67.2	76.4 **
Onset of sleep after midnight weekdays			41.2	48.6 *				
Sleep duration weekdays (hours)	6.9 ± 1.2	6.6 ± 1.2 ***			6.9 ± 1.1	6.6 ± 1.2 ***		

M ± SD = 0—mean value ± standard deviation of the parameter in those without complaints; M ± SD = 1—mean value ± standard deviation of the parameter in those with complaints; % = 0—percentage of those without complaints who had the parameter; % = 1—percentage of those with complaints who had the parameter; * *p* < 0.05; ** *p* < 0.01; *** *p* < 0.001.

**Table 3 ijerph-20-03309-t003:** Comparison of lifestyle parameters, health aspects and computer workstation features in subgroups of people with and without complaints in the neck region or lumbar spine during the pandemic (only statistically significant differences are presented).

	Neck				Lumbar			
	M ± SD = 0	M ± SD = 1	% = 0	% = 1	M ± SD = 0	M ± SD =1	% = 0	% = 1
Moderate PA (4MET·minutes·days per week)					562.2 ± 837.2	757.9 ± 1870 *		
Walking (3.3MET·minutes·days per week)	563.7 ± 760.2	742.5 ± 1409 *						
Following WHO standards for PA			33.9	49.0 **			54.2	62.4 *
Section A (chair) (points)	3.8 ± 1.4	3.9 ± 1.5 **			3.6 ± 1.4	3.8 ± 1.5 *		
Section B (monitor + phone) (points)	2.3 ± 0.8	2.1 ± 0.8 **						
Section D (monitor + additional equipment) (points)	2.8 ± 1.0	2.9 ± 1.2 *						
ROSA (final score) (points)	3.8 ± 1.4	4.0 ± 1.5 **			3.8 ± 1.4	4.0 ± 1.4 **		
Stress (moderate or higher)			79.0	86.0 *			79.7	87.0 **
Headache (yes)			73.0	86.6 ***			75.1	85.0 **

M ± SD = 0—mean value ± standard deviation of the parameter in those without complaints; M ± SD = 1—mean value ± standard deviation of the parameter in those with complaints; % = 0—percentage of those without complaints who had the parameter; % = 1—percentage of those with complaints who had the parameter; * *p* < 0.05; ** *p* < 0.01; *** *p* < 0.001.

## Data Availability

Not applicable.

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
