# Peer review of "Ergonomics of E-Learning Workstations and the Prevalence of Musculoskeletal Disorders—Study among University Students"

_ijerph, 2023, doi:10.3390/ijerph20043309_

Round 1

Reviewer 1 Report

The work of Janc and colleagues highlighted how the recent COVID-19 pandemic was responsible for an increase in musculoskeletal disorders induced by ergonomic deficiencies of e-learning stations required during periods of isolation.

The manuscript is well structured; however, it would be helpful to understand better how the authors retrieved the pre-COVID-19 data. Are they the result of previous surveys and questionnaires (if so, please indicate the reference period), or are they based on participants' 'memories'? In the latter case, it would be necessary to assess the possible bias induced by possible alterations in the perception of the activities carried out and attributable to the phenomena of social isolation

Author Response

Dear Reviewer,

Thank you for all your work on our manuscript “Ergonomics of e-learning workstations and the prevalence of musculoskeletal disorders – study among University Students in Poland” . In accordance with your recommendations we have made relevant corrections. Now, we are resubmitting the revised paper simultaneously providing a point-by-point response in attached document.

Reviewer 2 Report

General point:

This study aimed to evaluate the conditions of e-learning and the impact of the learning modality on the occurrence of musculoskeletal symptoms among university students in Poland, before the COVID-19 pandemic and during the period from October 2020 to June 2021. For this, were obtained information about lifestyle, ergonomic of computer workstations, the incidence and severity of musculoskeletal symptoms and headaches. The authors verified that during the COVID-19 pandemic, there was significant increase in musculoskeletal disorders and their intensity. This study is pertinent, despite the obviousness of the results. However, this study reports important ergonomic concerns to be investigated in different life situations.

Specific points:

Abstract: The abstract needs to be improved. In general, it does not present statistical information and an objective conclusion in relation to the observed comparative results.

Page 01, line 21: “modified IPAQ” If scientifically validated, indicate the year.

Page 01, line 22: “by ROSA method” Please, indicate the year.

Page 01, line 23: What is NMQ?

Page 01, lines 23-25: What was the statistical treatment adopted? Are these two groups independent and/or dependent? What the intensity? Please include this information in the abstract?

Page 01, line 25: Adopt 0.05.

Page 01, lines 25-26: The lack of ergonomic remote learning stations was evaluated or this is a hypothesis?

Keywords: Please do not use words that already appear in the title. Also, write the words in alphabetical order.

Introduction: The aim of this study was to evaluate e-learning conditions and the impact of the learning modality on the occurrence of musculoskeletal symptoms in university students from Poland (see title)? Is it possible to reproduce the results for other students? If the answer is positive, it is suggested to suppress the Polish information in the title.

Page 02, lines 57-58: “impact of the learning modality on the occurrence of musculoskeletal symptoms” Has this not been investigated in other studies? Highlight the originality of this study. Note: Please include a final punctuation in this sentence.

Page 02, line 92: “October 2020 to June 2021” When did the home restriction start? That is, what is the average time of home restriction? The restriction time can impact on neuromuscular disorders.

Page 02, line 93: What is WHO?

Page 03, lines 112-114: “In the analysis using the ROSA method, the period before the pandemic was omitted, due to the lack of a classic computer station during desktop learning resulting from the diversity of activities.” So, the ROSA method corresponds to a justification of the observed changes between periods, correct? Thus, the independent variable is the period (before and during COVID-19) and the dependent variables, the musculoskeletal disorders.

Page 03, line 136: “p<0.05” In the abstract was adopted 0.01. Adopt 0.05 throughout the manuscript.

Page 04, Table 1: Replace this table 1 by a sentence (adopt mean±DP). The min and max values are not necessary.

Page 04, lines 155-157, 165, 171-172, etc.: “p<0.01” Adopt p<0.05. Please, check the whole article.

Page 04, lines 156, 160, 164: Adopt mean±DP and one decimal to DP (p. e., 5.23 to 5.4).

Pages 04, 06-07: Tables 2, 3 and 4: Adjust the format and adopt an asterisk when p<0.05. Include the corresponding units of measurement that are missing.

Discussion: Relevant.

Conclusions: The conclusion (one conclusion) is in agreement with the verified results.

Author Response

(The authors gave the same response as above.)

Round 2

Reviewer 2 Report

This study aimed to evaluate the conditions of e-learning and the impact of the learning modality on the occurrence of musculoskeletal symptoms among university students, before the COVID-19 pandemic and during the period from October 2020 to June 2021. For this, were obtained information about lifestyle, ergonomic of computer workstations, the incidence and severity of musculoskeletal symptoms and headaches. The authors verified that during the COVID-19 pandemic, there was significant increase in musculoskeletal disorders and their intensity. This study is pertinent and, the changes made have substantially improved the manuscript. So, I recommend this manuscript for publication in International Journal of Environmental Research and Public Health.

So, I suggest adopting 0.05 in all manuscript and, don’t use words that already appear in the title.